DATA RELEASE

# Genome assembly and annotation of the king ratsnake, *Elaphe carinata*

Jiale Fan[1,2,3,†], Ruyi Huang[1,4,5,†], Diancheng Yang[1,5], Yanan Gong[1,5], Zhangbo Cui[1,5], Xinge Wang[1,5], Zicheng Su[1,5], Jing Yu[1,5], Yi Zhang[1,5], Tierui Zhang[1,5], Zhihao Jiang[1,5], Tianming Lan[2,6], He Wang[3,*] and Song Huang[1,5,*]

1 Anhui Province Key Laboratory of the Conservation and Exploitation of Biological Resource, College of Life Sciences, Anhui Normal University, Wuhu, 241000, China
2 State Key Laboratory of Agricultural Genomics, BGI-Shenzhen, Shenzhen, 518083, China
3 College of Wildlife and Protected Area, Northeast Forestry University, Harbin, 150040, China
4 Shanghai Collaborative Innovation for Aquatic Animal Genetics and Breeding, Shanghai Ocean University, Shanghai, 201306, China
5 Huangshan Noah Biodiversity Institute, Huangshan, 245000, China
6 BGI Life Science Joint Research Center, Northeast Forestry University, Harbin, 150040, China

## ABSTRACT

The king ratsnake (*Elaphe carinata*) of the genus Elaphe is a common large, non-venomous snake widely distributed in Southeast and East Asia. It is an economically important farmed species. As a non-venomous snake, the king ratsnake predates venomous snakes, such as cobras and pit vipers. However, the immune and digestive mechanisms of the king ratsnake remain unclear. Despite their economic and research importance, we lack genomic resources that would benefit toxicology, phylogeography, and immunogenetics studies. Here, we used single-tube long fragment read sequencing to generate the first contiguous genome of a king ratsnake from Huangshan City, Anhui province, China. The genome size is 1.56 GB with a scaffold N50 of 6.53M. The total length of the genome is approximately 621 Mb, while the repeat content is 42.26%. Additionally, we predicted 22,339 protein-coding genes, including 22,065 with functional annotations. Our genome is a potentially useful addition to those available for snakes.

**Subjects** Genetics and Genomics, Zoology, Animal Genetics

**Submitted:** 23 May 2023

\* Corresponding authors. E-mail: wanghenefu@163.com; snakeman@ahnu.edu.cn

† Contributed equally.

Preprint submitted at https://doi.org/10.20944/preprints202308.1797.v1

Included in the series: *Snake Genomes* (https://doi.org/10.46471/GIGABYTE_SERIES_0004)

## DATA DESCRIPTION

The king ratsnake (*Elaphe carinata*) belongs to the family Colubridae and the genus *Elaphe*. It is a large oviparous snake [1] found in many provinces of South-eastern China. The southern edge of its distribution area can reach northern Guangdong, Guangxi, and Taiwan, while the northern edge is in the Beijing-Tianjin area (Figure 1). The king ratsnake is also found in northern Vietnam and several Japanese islands (Ryukyu Islands, including the Senkaku Islands) [2, 3].

 *E. carinata* mainly inhabits mountainous and hilly areas, and generally feeds on rodents, birds, and eggs. Its juveniles differ significantly from adults in color and size. When threatened, *E.carinata* can use its anal glands to secrete a foul-smelling fluid [3]. King ratsnakes are farmed in many countries as an important food source as they provide a large amount of proteins [4]. According to the China Red Data Book of Endangered

**Figure 1.** An *E. carinate* individual photographed by Diancheng Yang.

Animals [5], the king ratsnake is listed as a vulnerable species. The common name "king ratsnake" refers to its habit of eating other snakes, thanks to a unique protein in its blood. The non-venomous king snake exhibits a strong antagonistic effect against the venom of various poisonous snakes, including those with blood-circulating poisons (such as the bamboo leaf green snake (*Trimeresurus stejnegeri*) and the sharp-nosed viper (*Deinagkistrodon acutus)*) and neurotoxins (such as the many-banded krait (*Bungarus multicinctus*), one of the most lethal snakes in the world). However, the exact immune mechanism for this protection and the pathways for digesting these poisons are unknown.

The development of genome research technology has advanced the research of reptile evolution, including the origin and production of snakes and their toxins [6, 7]. However, limited research has been dedicated to the natural antivenoms of snakes. As snake antivenoms are the only treatments for effectively preventing or reversing the effects of snake venoms [8], the genome of the king ratsnake may provide new insight into antivenoms and aid in the study of its digestive mechanisms.

In the present study, we assembled the first highly contiguous *E. carinate* genome using single-tube long fragment read (stLFR) sequencing data combined with next-generation sequencing for gap filling and redundant contigs removal. The resulting genome, comparable in size to a previously sequenced corn snake *Pantherophis guttatus* [9] but more contiguous, is a valuable resource for future studies. For instance, it could support studies on snake evolution and venom immunity.

## MAIN CONTENT
### Context

As a snake with a long history of captive breeding, the reproduction and the viruses carried by the king ratsnake have been well studied [10, 11]. However, there is insufficient research on its immune resistance and a general lack of genomic resources. Here, we provide the *de novo* assembly of a highly contiguous king ratsnake genome with a size of 1.56 Gb based on stLFR sequencing data. The maximal scaffold length is 49.75M, the scaffold N50 length is 6.53M, and the contig N50 is 44.05 Kb, with a GC content of 40.25%. Compared to many other published snake genome sequences, the genome we assembled is highly contiguous. Our draft genome sequence of *E. carinata* will be an invaluable resource for understanding snake venom resistance.



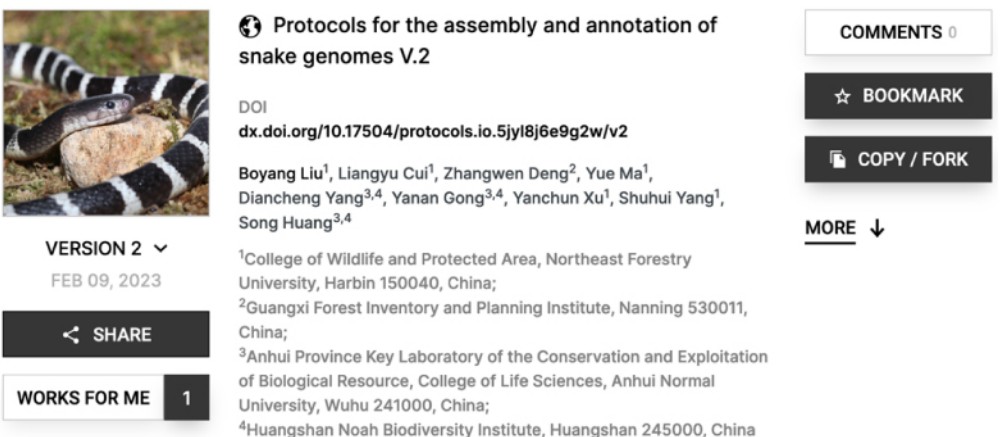

**Figure 2.** A protocols.io collection of protocols for sequencing snake genomes [12]. https://www.protocols.io/widgets/doi?uri=dx.doi.org/10.17504/protocols.io.5jyl8j6e9g2w/v2

## Methods

Experimental procedures used in this study and more detailed methods are available via a protocol collection hosted in protocols.io (Figure 2) [12].

## Samples and ethics statement

An adult *E.carinata* (NCBI:txid74364) individual from Huangshan City in the Anhui province was collected for DNA and RNA sequencing. After the individual died naturally, the samples were transferred to dry ice, quickly frozen, and kept at −80 °C until further use. For RNA sequencing, we used tissues from four organs: liver, stomach, kidney, and muscle. However, we performed stLFR sequencing using muscle samples only. Sample collection and experimental studies were both approved by the Institutional Review Board of BGI (BGI-IRB E22017). All procedures were carried out following the guidelines of the BGI-IRB.

## Nucleic acid isolation, library preparation, and sequencing

We extracted DNA according to the method described by Wang *et al.* [10]. An stLFR co-barcoded DNA library was constructed using the MGIEasy stLFR Library Prep Kit (MGI, China). Sequencing was performed using a BGISEQ-500 sequencer. The genomic DNA kit (AxyPrep, USA) was used to isolate DNA for whole-genome sequencing (WGS). Total RNA was extracted according to the manufacturer's instructions using the TRlzol reagent (Invitrogen, USA). The integrity and concentration of DNA and RNA samples were assessed using a Qubit 3.0 Fluorometer (Life Technologies, USA) and Agilent 2100 Bioanalyzer System (Agilent, USA). Finally, we used 200–400 bp RNA fragments for reverse transcription of cDNA libraries (Table 1).

## Genome assembly, annotation, and assessment

The stLFR sequencing data were assembled using Supernova software (v2.1.1, RRID:SCR_016756) [13]. Based on the WGS data, the assembly was gap-filled using GapCloser (v1.12-r6, RRID:SCR_017633) [14], and redundant contigs were removed using redundans (v0.14a) [15].

**Table 1.** Summary of our sequencing data of *E. carinata*.

|  | stLFR | WGS | RNA-seq |
|---|---|---|---|
| Sam ReadNum (M) | 909.19 | 530.66 | 56.94 |
| Sam BaseNum (Gb) | 209.11 | 106.14 | 11.38 |
| Sam GC (%) | 40.76 | 40.48 | 44.21 |
| Sequencing Depth (×) | 134 | 68 | N/A |

We first identified *de novo* repeats using Tandem Repeat Finder [16] (v. 4.09), LTR_finder (v1.0.6, RRID:SCR_015247) [17], and RepeatModeler [18] (v1.0.8, RRID:SCR_015027). These repeats were then used together with Repbase (RRID:SCR_021169) in RepeatMasker [19] (v. 3.3.0, RRID:SCR_012954) as known elements for identifying transposable elements. Also, known repeat elements were searched using RepeatProteinMask [20] (v. 3.3.0) in genome sequences. For *de novo* protein-coding gene prediction, we first used Augustus [21] (v3.0.3, RRID:SCR_008417). Based on the RNA-seq data filtered clean by Trimmomatic [22] (v0.30, RRID:SCR_011848), the transcripts were assembled using Trinity [23] (v2.13.2, RRID:SCR_013048), and compared with the king ratsnake genome through Program to Assemble Spliced Alignments (or PASA) [24] (v2.0.2, RRID:SCR_014656) to obtain the gene structures. For homology-based prediction, we used Blastall [25] (v2.2.26) with an *E*-value cut-off of $1 \times 10^{-5}$ to map the protein sequences by comparing our king ratsnake genome with four high-quality data of *Crotalus tigris*, *Pseudonaja textilis*, *Notechis scutatus*, and *Thamnophis elegans* from the UniProt database (release-2020_05, RRID:SCR_004426). GeneWise [26] (v2.4.1, RRID:SCR_015054) was used to analyze alignment results to predict gene models. We used the MAKER pipeline [27] (v3.01.03, RRID:SCR_005309) to generate the final gene set representing RNA-seq, homology, and *de novo* predicted genes.

To perform BLAST comparisons on structurally annotated gene sets, functional annotation was completed using SwissProt [28], TrEMBL [28], and Kyoto Encyclopedia of Genes and Genomes (KEGG; RRID:SCR_012773) [29] databases, with the *E*-value cut-off set to $1 \times 10^{-5}$. InterProScan [30] (v5.52-86.0, RRID:SCR_005829) was used to count and visualize structural domain information, and Gene Ontology (GO; RRID:SCR_002811) terms were used for gene enrichment.

The genome integrity was evaluated using Benchmarking Universal Single-Copy Orthologs (or BUSCO, v5.2.2, RRID:SCR_015008), with parameters set to genome-mode and dataset input set to vertebrata_odb10 [31].

A maximum-likelihood tree was built using OrthoFinder (v2.3.7, RRID:SCR_017118) [32] with default parameters based on the protein sequences of *Rana temporaria* (GCA_905171775.1), *Gopherus evgoodei* (GCA_007399415.1), *Podarcis muralis* (GCA_004329235.1), *P. textilis* (GCA_900518735.1), *T. elegans* (GCA_009769535.1), *P. guttatus* (GCA_001185365.2), and *B. multicinctus* (GWHBJIQ00000000), all of which are species having close genetic relationships with *E. carinata*. Specifically, protein sequences were used to infer orthogroups. Then, an unrooted gene tree was inferred for each orthogroup. Specifically, the unrooted species tree was inferred from unrooted gene trees using the STAG algorithm [33] and finally rooted using the STRIDE algorithm [34].

**Table 2.** Summary of the features of our *E. carinata* genome.

| | Contig | Scaffold |
|---|---|---|
| Maximal length (bp) | 657,733 | 52,164,798 |
| N90 (bp) | 3,039 | 4,090 |
| N50 (bp) | 45,108 | 6,847,971 |
| Number ≥ 500 bp | 187,253 | 134,573 |
| Ratio of Ns | 0.059 | 0.059 |
| GC content (%) | 40.25 | 40.25 |
| Genome size (bp) | 1,574,091,846 | 1,674,021,862 |

**Table 3.** Content of various repeat sequences in our *E. carinata* genome.

| Type | Length (Bp) | % in genome |
|---|---|---|
| DNA | 36,972,435 | 2.208599 |
| LINE | 560,947,698 | 33.508983 |
| SINE | 10,266,832 | 0.613303 |
| LTR | 147,804,256 | 8.829291 |
| Other | 0 | 0 |
| Satellite | 1,783,176 | 0.10652 |
| Simple repeat | 8,313,568 | 0.496622 |
| Unknown | 6,954,098 | 0.415413 |
| Total | 707,433,803 | 42.259532 |

**Table 4.** Summary of TEs in our *E. carinata* genome.

| | Repbase TEs | | TE proteins | | *De novo* | | Combined TEs | |
|---|---|---|---|---|---|---|---|---|
| Type | Length (Bp) | % in genome | Length (Bp) | % in genome | Length (Bp) | % in genome | Length (Bp) | % in genome |
| DNA | 44,586,593 | 2.663442 | 3,037,369 | 0.181441 | 114,900,759 | 6.863755 | 137,315,177 | 8.202711 |
| LINE | 172,974,640 | 10.332878 | 142,896,461 | 8.536117 | 257,937,611 | 15.408258 | 287,262,246 | 17.160006 |
| SINE | 27,330,057 | 1.632599 | 0 | 0 | 42,327,923 | 2.528517 | 52,336,172 | 3.126373 |
| LTR | 20,332,067 | 1.214564 | 26,146,398 | 1.561891 | 36,199,886 | 2.16245 | 48,061,022 | 2.870991 |
| Other | 2,8331 | 0.001692 | 291 | 0.000017 | 0 | 0 | 28,622 | 0.00171 |
| Unknown | 0 | 0 | 0 | 0 | 214,450,953 | 12.810523 | 214,450,953 | 12.810523 |
| Total | 252,872,307 | 15.105675 | 171,980,912 | 10.273516 | 645,389,076 | 38.553205 | 685,733,449 | 40.963231 |

## RESULTS

### Genome

StLFR was used to generate the *E.carinata* genome assembly as it is a fast and cost-effective sequencing technology. After gap filling and redundant contigs removal, the total size of the genome assembly is 1.67 Gbp (Table 2).

Usually, genome-wide repetitive elements are important for eukaryotic evolution [35]. In *E. carinata*, the content of repetitive elements in the genome accounted for 42.26%, and the total length reached 621 Mb (Tables 3 and 4). Among all repetitive elements, long interspersed nuclear elements (LINEs) accounted for 72.56%, DNA for 4.78%, and unknown types for 0.90% (Figure 3). This indicates that the content and quantity of repetitive elements is one of the sources of species differences.

### Annotation

A total of 22,065 functional genes were annotated, and the annotations associated with the TrEMBL database accounted for the most significant proportion, reaching 97.92% (Table 5). In addition, all genes were annotated with KEGG, which showed the highest number in pathways such as Human Diseases, Organismal Systems, and Metabolism, while the highest

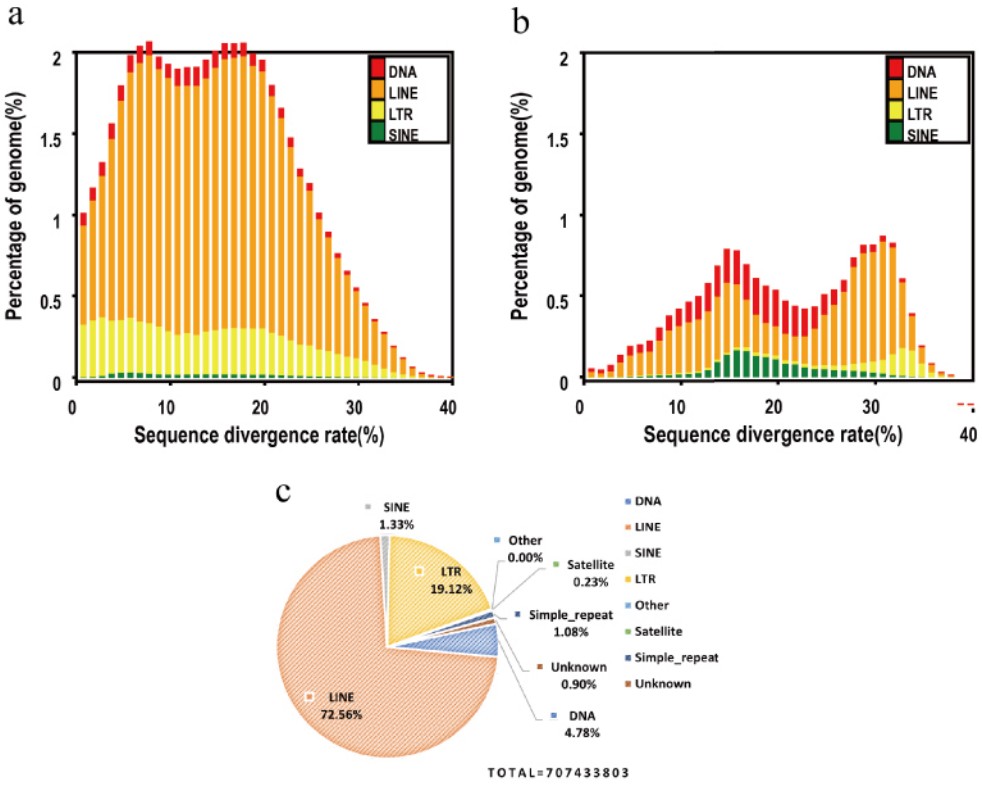

**Figure 3. Distribution of transposable elements (TEs), such as DNA transposons (DNA) and RNA transposons in our *E. carinata* genome. RNA transposons include DNAs, LINEs, long terminal repeats (LTRs), and short interspersed nuclear elements (SINEs).**
(a) Distribution of divergence rates for *de novo* sequences. (b) Distribution of divergence rates for known sequences. (c) Proportion and distribution of repeating elements.

**Table 5.** Summary of the annotation results in our *E. carinata* genome.

| Values | Swissprot-Annotated | KEGG-Annotated | TrEMBL-Annotated | Interpro-Annotated | GO-Annotated | Overall |
|---|---|---|---|---|---|---|
| Number | 20,796 | 19,836 | 21,874 | 21,604 | 15,169 | 22,065 |
| Percentage | 93.09% | 88.80% | 97.92% | 96.71% | 67.90% | 98.77% |
| Total number of gene | | | 22,339 | | | |
| Average cds length | | | 1,292.780295 | | | |
| Average exon length | | | 181.041757 | | | |
| Average mRNA length | | | 20,146.9872868078 | | | |
| Total number of exons | | | 159,518 | | | |

number of Signal Transduction genes were found in Environmental Information Processing. Additionally, GO gene annotation of *E. carinata* revealed that, among 25 biological process pathways, 251 genes were related to immune system processes, and two genes were related to detoxification (Figure 4).

## DATA VALIDATION AND QUALITY CONTROL

When assessing the quality of the genome, we performed a completeness assessment of the assembly with BUSCO v3.1.0 [36] using the vertebrata_odb10 dataset [31]. This assembly matched 83.2% of the complete BUSCOs (Figure 5).



Figure 4. **Gene annotation results for *E. carinata*.**
(a) KEGG annotation of *E. carinata*. (b) GO enrichment of *E. carinata*.

By screening closely related species, *R. temporaria*, *G. evgoodei*, *P. muralis*, *P. textilis*, *T. elegans*, *P. guttatus*, and *B. multicinctus* were filtered to construct a phylogenetic tree.

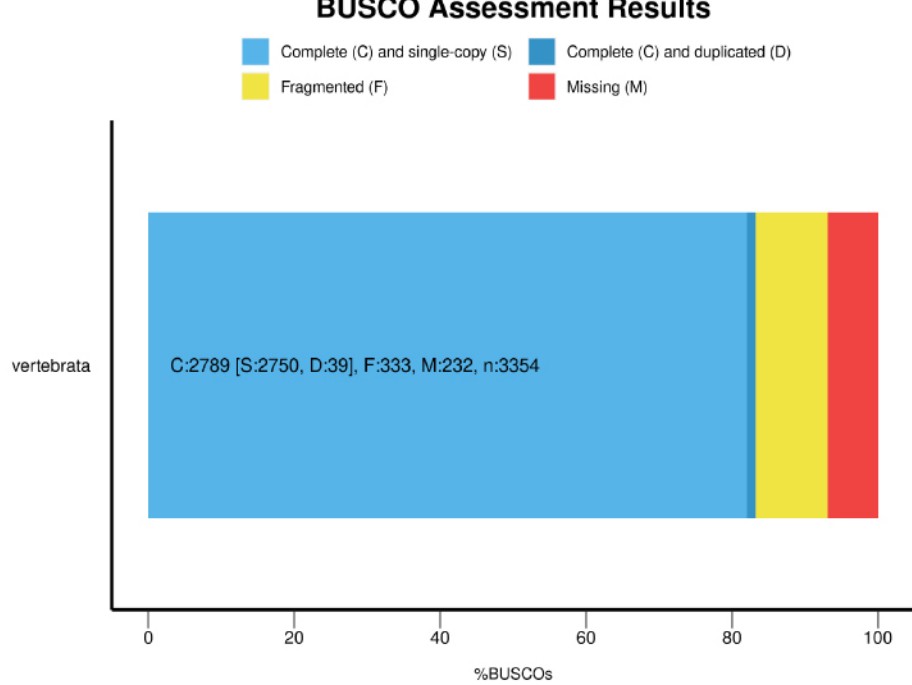

**Figure 5.** BUSCO assessment result of the *E. carinata* genome.

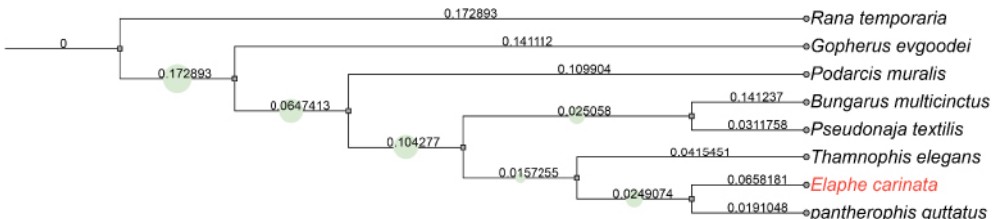

**Figure 6. Maximum-likelihood tree reconstructed using protein sequences.**
The numbers represent the branch lengths. The colored squares represent bootstraps/metadata from 0.310873 to 1.

Among the 288,969 genes, 280,103 (96.9% of the total) were assigned to 22,072 orthogroups. Consistently with previous studies [37], our data can construct phylogenetic trees and cluster closely related species (Figure 6).

## REUSE POTENTIAL

The king ratsnake has both nutritive and medicinal value, and the growth and development of individuals and snake eggs have been widely studied [38]. However, there are insufficient studies and genomics data on its immune system. Only Sun et al. researched the development of the immune system during the embryonic stage of the king snake [39].

Our data can be combined with other snake genome data for phylogenetic studies to construct the developmental evolutionary history of snakes and other reptiles. In addition, our genomic data can provide new insights into the study of the immune system, snake venom resistance genes, and their mechanisms of action.

## DATA AVAILABILITY

The data that support the findings of this study have been deposited into the CNGB Sequence Archive (or CNSA) [40] of the China National GeneBank DataBase (or CNGBdb) [41] with the accession number CNP0004039. Raw sequencing data is also in the SRA under bioproject PRJNA955401, and additional data is available in the GigaDB repository [42].

## EDITOR'S NOTE

This paper is part of a series of Data Release papers presenting the genomes of different snake species [43].

## ABBREVIATIONS

GO, Gene Ontology; KEGG, Kyoto Encyclopedia of Genes and Genomes; LINES, long interspersed nuclear elements; LTRs, long terminal repeats; SINE, short interspersed nuclear elements; stLFR, single-tube long fragment read; TEs, transposable elements; WGS, whole-genome sequencing.

## DECLARATIONS

### Ethics approval and consent to participate

Sample collection and experimental studies were both approved by the Institutional Review Board of BGI (BGI-IRB E22017).

### Competing interests

The authors declare no conflict of financial interests.

### Authors' contributions

SH, HW, and TL designed and initiated the project. YZ, TZ, ZJ, and JY collected the samples. XW and ZS performed the DNA extraction. DY, YG, and ZC generated the genome assembly. JF and RH performed the data analysis and wrote the manuscript. All authors read and approved the final manuscript.

### Funding

Our project was financially supported by the Doctoral Research Starting Foundation of Anhui Normal University (752017), the National Natural Science Foundation of China (NSFC 31471968), and Guangdong Provincial Key Laboratory of Genome Read and Write (grant no. 2017B030301011). This work was also supported by China National GeneBank (CNGB).

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
