## [Reviewer Report]

Comments on revised manuscriptAuthors have answered all my concerns very well. I have no further requirments.

---

## [Editor Report]

Editor’s AssessmentThe king ratsnake Elaphe carinata is a common large non-venomous snake that is widely distributed in Southeast and East Asia,. As a non-venomous snake, the king snake that is predatory on venomous snakes. To better understand the immune mechanisms of how it evades venom, a 1.56Gb reference genome was sequenced, assembled and described in this work.. With other snake species already published, this data can be combined with these and other upcoming snake genome data to construct the evolutionary history of snakes and other reptiles as well as the genetic basis of venom evasion.

---

## [Reviewer Report]

Reviewer name and names of any other individual's who aided in reviewer Jun LiDo you understand and agree to our policy of having open and named reviews, and having your review included with the published papers. (If no, please inform the editor that you cannot review this manuscript.)YesIs the language of sufficient quality?YesPlease add additional comments on language quality to clarify if needed
Are all data available and do they match the descriptions in the paper? YesAdditional CommentsAre the data and metadata consistent with relevant minimum information or reporting standards? See GigaDB checklists for examples <a href="http://gigadb.org/site/guide" target="_blank">http://gigadb.org/site/guide</a>YesAdditional CommentsIs the data acquisition clear, complete and methodologically sound?NoAdditional CommentsThe method is incomplete, readers will be hard to assess the quality of the genome. For example, authors did not report their sequencing depth for genome assembly, which is a important information for evaluating the sufficiency of data and thus the accuracy of the assembly. I suggest providing a summary table for the total data generated from different sequencing methods and their corresponding sequencing depth (eg. how many data generated from the stLFR and from the next-generation sequencing?). Is there sufficient detail in the methods and data-processing steps to allow reproduction?YesAdditional CommentsIs there sufficient data validation and statistical analyses of data quality? YesAdditional CommentsIs the validation suitable for this type of data?YesAdditional CommentsIs there sufficient information for others to reuse this dataset or integrate it with other data?YesAdditional CommentsAny Additional Overall Comments to the AuthorThis manuscript provide a genome resource of the king ratsnake. Since the title said “helps reveal its biological characteristics”, authors should at least provide some explanation or discussion on this aspect. The total structure of this MS is not classic. For example, the “Reuse potential” section after the Results. Is it acquired for a database type article in this journal? If not, I suggest providing a Discussion section.   Minor comments: Line 31. Change “that is predatory on” to “can prey on”.  Line 32. This sentence has a grammar error, please explain “of which” here. Except for the immune specificity, the digestion on venomous snakes are also important for this species.  Line 36. Is it a real “complete” genome?  Lines 57-58. This sentence is ambiguous, is this trait of secreting fluid when threatened specific for juveniles or adults?  Line 155. Why choose these species to construct phylogeny?  Lines 159-160. What software was used to construct the phylogenetic tree?  Line 164. Since the assembled genome still has gaps (Ns), you can not say it “a complete genome” in the abstract (line 36).  Lines 169-170. The unknown type of repetitive sequences accounts for the second largest component of all repeats. Is it reasonable? What about other published snake genomes? If other published snake genome also showed high proportion of unknown repeats, it may indicate that the current repetitive sequence database or methods are unsuitable for predicting snake genomes. But if not, I would suspect the accuracy of your result.  Line 176. The species name show be italic even in the legend of figure.  Lines 194, 204. I guess these are not “enrichment”, but just an “annotation”. GO and KEGG Enrichments mean that you have a focused gene sets and want to know their enriched GO terms and KEGG pathways. Here you just annotated all genes into these database and checked their assignments. Besides, the font size in figure 4 is too small to read. Please adjust it.  Line 223. What do the numbers one the branch lines mean? Please clarify. Also, the species names should be italic.  Line 233. Change ‘done’ to ‘conducted’.RecommendationMinor Revision

---

## [Reviewer Report]

Reviewer name and names of any other individual's who aided in reviewer Xu JiangDo you understand and agree to our policy of having open and named reviews, and having your review included with the published papers. (If no, please inform the editor that you cannot review this manuscript.)YesIs the language of sufficient quality?NoPlease add additional comments on language quality to clarify if needed
Are all data available and do they match the descriptions in the paper? YesAdditional CommentsAre the data and metadata consistent with relevant minimum information or reporting standards? See GigaDB checklists for examples <a href="http://gigadb.org/site/guide" target="_blank">http://gigadb.org/site/guide</a>YesAdditional CommentsIs the data acquisition clear, complete and methodologically sound?YesAdditional CommentsIs there sufficient detail in the methods and data-processing steps to allow reproduction?YesAdditional CommentsIs there sufficient data validation and statistical analyses of data quality? NoAdditional CommentsIs the validation suitable for this type of data?NoAdditional CommentsIs there sufficient information for others to reuse this dataset or integrate it with other data?YesAdditional CommentsAny Additional Overall Comments to the AuthorKing snake is a common large non-venomous snake with obviously economical and medicinal value. The stLFR is a fast and cheap tools for animal whole genome assembly. Therefore the draft map of kingsnake using stLFR is a good source for further use for the field of reptile research. Before futher evaluation, there are some issues should be addressed: 1. Line 72, the "combined with ngs data for correction" is not need here. Because data correction is a common standard for stLFR assembling process. 2. In Data description part and phylogenetic part(Fig.6), as authors discussed the antivenome mechanism, I think Bungarus multicinctus is a good object to compare and discussion. The B.multicinctus is the most lethal snake in china and had habit overlap with king snake. For phylogenetic analysis, the elapids are related with colubrids. There are some genomic works of B.multicinctus(doi: 10.1186/s13020-021-00502-6. ; doi: 10.1016/j.apsb.2022.11.015.).  3. In context part, it had better to add the contig N50 result. 4. For Fig2, it is suggested to remove the name at the bottom of the figure. 5. In table4, it is suggested to add some characteristics of the predicted genes as the mean length of genes, mean length of CDS and the exon numbers etc.RecommendationMajor Revision